# Pleiotrophin Interaction with Synthetic Glycosaminoglycan Mimetics

**DOI:** 10.3390/ph15050496

**Published:** 2022-04-19

**Authors:** Jonathan R. Miles, Xu Wang, Jose L. de Paz, Pedro M. Nieto

**Affiliations:** 1Glycosystems Laboratory, Instituto de Investigaciones Químicas (IIQ), cicCartuja, CSIC and Universidad de Sevilla, C/Américo Vespucio, 49, 41092 Sevilla, Spain; jonathan.miles@nottingham.ac.uk (J.R.M.); jlpaz@iiq.csic.es (J.L.d.P.); 2School of Molecular Sciences, Arizona State University, Tempe, AZ 85281, USA; xuwang@asu.edu

**Keywords:** glycosaminoglycan, chondroitin sulphate type E, NMR, ^15^N-HSQC

## Abstract

Chondroitin sulfate (CS) E is the natural ligand for pleiotrophin (PTN) in the central nervous system (CNS) of the embryo. Some structures of PTN in solution have been solved, but no precise location of the binding site has been reported yet. Using ^15^N-labelled PTN and HSQC NMR experiments, we studied the interactions with a synthetic CS-E tetrasaccharide corresponding to the minimum binding sequence. The results agree with the data for larger GAG (glycosaminoglycans) sequences and confirm our hypothesis that a synthetic tetrasaccharide is long enough to fully interact with PTN. We hypothesize that the central region of PTN is an intrinsically disordered region (IDR) and could modify its properties upon binding. The second tetrasaccharide has two benzyl groups and shows similar effects on PTN. Finally, the last measured compound aggregated but beforehand, showed a behavior compatible with a slow exchange in the NMR time scale. We propose the same binding site and mode for the tetrasaccharides with and without benzyl groups.

## 1. Introduction

The neurotrophic and developmental factor pleiotrophin (PTN), together with midkine (MK), constitute the neurite growth-promoting factor family [1]. They are potent cytokines that act through proteoglycan membrane receptors modulating their activity that is associated with highly sulfated GAG interaction, mainly CS type E or heparin. The main receptor for PTN is the transmembrane-based tyrosine kinase receptor-type protein tyrosine phosphatase ζ (PTPRZ). PTNs main expression occurs during CNS embryo formation but participates in other physiological processes such as tissue repair, cell differentiation, inflammation, angiogenesis, and cancer [2]. 

GAG belongs to a family of extracellular linear sulfated polysaccharides that includes heparin and chondroitin sulfate (CS). Their basic structure is composed of a sulfated repeating disaccharide formed by a hexosamine and a uronic acid. Their unusual biosyntheses yield long and complex chains, the primary source of heterogeneity is the diverse sulfation patterns over the same chain. GAG regulates many biological processes via the interaction with a large number of proteins [3]. 

The neurite growth-promoting factor family can be considered a relevant molecular target for the treatment of various diseases. Therefore, there is a great interest in the discovery of inhibitors. It is known that MK and PTN strongly bind to heparin and CS chains and that these molecular recognition events are essential for protein activity [4]. From the small molecule perspective, we studied the binding of synthetic tetrasaccharides with MK and PTN using NMR transient techniques, transfer NOE and STD, and molecular modeling [5,6,7]. PTN solution structure and several complexes were described by one of us using NMR methods. From the solution structure it is relatively straightforward to analyze the role of the GAG substitution in the interaction using ^15^N-HSQC titrations [8,9]. In this work, we studied the interaction between PTN and ligands **1**–**3** from the protein perspective, applying the same methods as previously [8,9] to explore the minimum length for interaction together with the role of the hydrophobicity.

The synthetic tetrasaccharides **1**–**3** are pure compounds corresponding to the formulae in Figure 1 [6]. Formula **1** is a model of CS-E sequence, except having the sulfate in position 4 at the non-reducing terminal and the 4-methoxyphenyl group at the reducing-end to ensure consistency of the alpha-beta stereochemistry. Formula **2** has two additional benzyl groups in position 3 of the glucuronate rings, and **3** is fully O-substituted.

Synthetic methods have several advantages compared with the other principal source of glycosaminoglycans, which is the hydrolysis of the natural polymers. The most obvious difference is the unsaturation created by the depolymerization reaction in the non-reducing end. This unsaturation makes it impossible to distinguish between GlcA and IdoA on the parent compound. At the other end, there is an unblocked sulfated galactosamine residue. The presence of both anomers, alpha, and beta, is another source of heterogeneity [10]. Therefore, the synthetic tetrasaccharides should be considered equivalent to dp6 from the binding viewpoint.

Their synthesis has been described, and their interaction with midkine has been analyzed using NMR transient methods [6]. Their interaction with PTN has also been investigated [5,7]. We have found that introducing aromatic moieties increases the affinity for both MK and PTN [5,6,11]. We decided to investigate the complex structure from the protein perspective to assess if the interaction of the mimetics occurs in the same site or at an alternative location.

To characterize the interaction between the saccharides and MK or PTN, we used fluorescence polarization competition assay, with a fluorescence probe prepared by us, monitoring the fluorescence polarization decrease with ligand addition [12].

The PTN structure has two pseudotrombospondin type 1 repeat domains (TSR) stabilized by disulfide bonds, linked by a central disordered region, and with both ends also unstructured. Basic amino acids predominate in both ends and both beta-sheet domains. They can be grouped in clusters. In the NTD, R35, R39, R49, R52, and K54 are in the same face, the internal one. The CTD residues K84, R86, K107, and K68, K91, and R92, classified as clusters 1 and 2, respectively, correspond to the region close to the central linker and the loop between strands 2 and 3 of CTD. In some of the structures deposited in the protein data bank clusters 1 and 2 are spatially close. There is another group of basic residues on CTD in strands 2 and 3, K84, R86, and K107, directed towards the internal part of PTN in some of the structures [9,13]. PTN is very flexible; the 10 structures that characterize the ensemble, compatible with the experimental results in the PDB database, are highly dispersed (2N6F) [9]. If a single TSR is used for superimposition, then a clearer superimposition can be observed. Then, the central connection between the NTD and CTD is at the origin of this behavior.

In this work, we studied the binding of CS-E tetrasaccharides **1** and two mimetics **2** and **3**, using protein NMR techniques, ^15^N-HSQC [12]. The data indicate that the perturbations induced by CS-E tetrasaccharide **1** are similar to the mimetic **2** with two benzylic substituents [11].

## 2. Results

We applied the chemical shift perturbation mapping technique to compare at atomic resolution the effects of **1**, **2**, and **3** on the interaction with PTN using ^15^N-HSQC experiments at different concentrations of ligands. This technique allows the interaction site to be defined. The initial assignment of the PTN ^15^N-HSQC in measuring buffer was carried out using the data from BMRB [14] (accession number 25762, see Appendix A). After this, a titration was performed, adding **1**, **2**, or **3** from molar ratios from 0.0 to 3.0–4.0. Along with the titration, chemical shift differential displacements were detected. Once processed, the experiment array was analyzed using the MBinding module of MNova for a 1:1 complex. Compounds **1** and **2** produced changes compatible with binding in a fast exchange regime in the NMR time scale and showed similar results to the previous cases described by Wang [9,13] (Figure 1).

In the case of **3,** the protein peaks disappeared along with the titration. As a result, the precise exchange regime is unclear; it may be the consequence of a combination of slow exchange followed by aggregation, unfolding or another process that dismisses the amount of observable protein in solution (Figure 2). A similar ligand-induced signal decrease was seen in the titration of PTN by heparin reported by Rauvala [13]. In both cases, the final results are similar, as shown in Figure 2, with signals that can be attributed to unfolded structures.

Figure 3 displays the CSP of **1** and **2** along the chain, and in the top are the secondary structure elements, beta-strands, and the NTD and CTD region indicated at the same scale. The results obtained for tetrasaccharides **1** and **2** were qualitatively similar to previous data on CS-E (dp6), CS-A (dp8) [9], heparin (dp6), CS-E (dp6), DS (dp8), and CS-A (dp8) [8]. 

The global CSP profile of both compounds is very similar. The magnitude of the variation is clearly more significant for **2**, likely because of the influence of the aromatic rings of the *O*-benzyl groups (see Figure 3). When the secondary structure is considered, there are two regions where the larger relative CSP values are concentrated: the hinge region and the connection between the 2nd and 3rd beta-strands of the CTD. These two regions are close together in the space in some of the structures from the Protein Data Bank (2n6f), see Figure 4. Interestingly, the CSP magnitudes are very different for both compounds that are larger for **2** reaching until approximately 0.6. While in the case of **1** the maximum is 0.28.

From the normalized curves CSP vs. ligand to protein ratio, binding constants were calculated for each residue assuming a 1:1 model (see Figure 5, Table 1, and Appendix A). This analysis contradicts the previous fluorescence polarization data with stronger binding for the tetrasaccharide **2** with two benzylic substituents for both MK and PTN [5,6].

## 3. Discussion

We used ^15^N-HSQC to analyze the binding of **1**–**3** oligosaccharides to ^15^N labeled PTN from the protein perspective. Formulae **1**–**3** are tetrasaccharide mimics of the CS-E sequence, with a variable number of aromatics substituents, one, three, and seven. We previously presented the concept of CS-E mimetic with improved affinity for MK and PTN by introducing aromatics rings in their structure [5]. Compounds **1** and **2** are in the fast exchange regime, and some peaks move along the titration (Figure 1). From the variation of CSP with ligand concentration, the binding constants were calculated for several individual residues. Almost all the backbone amide peaks can be observed and assigned, but not all showed changes in the chemical shifts. Compound **3** behaves differently; in the first titration points, the intensity of the signals decreases, until peaks disappear, but they do not reappear in the complex position as can be expected for a slow chemical shift equilibrium. As an alternative mechanism, aggregation or unfolding is a plausible hypothesis. 

The IC_50_ measured by fluorescence polarization competition assay for PTN were proportional to the number of aromatics rings in the molecule [5]. The K_D_ calculated from CSP data for each residue for a 1:1 complex follow the opposite trend, and their affinities are lower for **2**. A possible explanation can be that as concentrations are so different, the stoichiometry varies from the FPCA to the NMR data. 

The results here presented fully support, also from the protein perspective, that the binding of the CS-E tetrasaccharide and the mimetic with two benzyl moieties, **2**, occurs in the same region and with similar characteristics compared to **1** [5,8,9]. Therefore, **2** can be considered a mimetic of **1** (Figure 3). The more significant chemical shift perturbations for each complex analyzed individually are located in identical regions of the PTN: in the hinge region, R52–C68, between NTD, G16–R52, and CTD, C68–C109, and in CTD in the connecting the loop between strands 2 and 3 of the beta-sheet, R92–T102, (see Figure 1 and Figure 2). Both structural elements are close together in the space in some NMR structures, suggesting that this is a critical element in the formation of the complexes. When we analyzed the individual residues in depth as W, that used to be involved in interaction with sugars, we found that only one from four were classified as interacting residues. This can be explained by considering that ^15^N-HSQC analysis is directed towards the variation of the backbone chemical shifts, while the carbohydrate–Trp interaction is between side chains.

There is a discrepancy between the magnitude of the chemical shift changes and K_D,_ with the most currently accepted rule being that the magnitude is proportional to the strength of the interaction. In this case, the chemical shift variations are larger for **2** with two benzylic substituents, but the K_D_ are weaker than those estimated by fluorescence polarization at lower concentrations. This can be explained by the effect of the aromatic chemical shift ring currents. The standard chemical shift perturbation analysis reflects the impact of hydrogen bonding on the backbone-specific interactions with the ligands [15]. Therefore, this technique spots the binding regions in rigid proteins with preorganized binding sites. In the PTN case, the main chemical shift differences occur in non-organized regions: the hinge loop and the second loop of the last TRS domain. This can be attributed to small changes in the distribution of conformations, observed on the NMR structures but non-reflected by changes in the secondary structure. Interestingly, these variations are more significant when aromatic substituents are included, as in **2**. We interpreted this feature because introducing two aromatics rings in one of the ligands increments the effect in the ring current perturbation by the aromatic ring, but this is not reflected in the binding strength. This may also be the reason for the dispersion of the K_D_ values obtained from the titration curves per residue.

## 4. Materials and Methods

^15^N Pleiotrophin was obtained from bacterial expression as described in previous works using M9 media supplemented with ^15^NH_4_Cl using standard procedures for the expression and purification.

NMR experiments were recorded on a 600 Bruker Avance III instrument (Bruker BioSpin GmbH, Silberstreifen 4, D-76287 Rheinstetten Germany) operating at 600 MHz, equipped with a cryoprobe 600, QCI Cryo 5m (1H/19F 15N/13C) for 1H, 15N, 13C, and 19F with 2H decoupling, using samples with 0.5 mM ^15^N-pleiotrophin in 300 μL of buffer 10 mM MES, pH = 6.0, 150 mM NaCl. Pleiotrophin was obtained as described by Wang [9]. The NMR tubes used were 3 mm wide in the measurement region. The titration was performed using previously freeze-dried aliquots corresponding to the amount of saccharide and buffer needed at the next points, dissolving it in the previous sample and returning it to the same tube (Table 2). We conducted the following HSQC experiments for the three compounds using the pulse sequence hsqcetfpf3gpsi from the NMR Bruker library [16,17,18,19].

Spectra were processed with Bruker TopSpin, and the real part of the spectra was converted into MNova; alternatively, the spectra were directly processed with MNova. We did not observe significant differences. They were all processed with partial linear prediction in the indirect dimension.

The peak assignment was carried out using the chemical shift list deposited in the BMRB (Biological Magnetic Resonance Bank, 25762) [14].

Data were analyzed with MNova Binding module, obtaining the fitted titration curves used to obtain K_D_, CSP, and their respective errors for each residue assigned, according to the manual [20].

## 5. Conclusions

This work completes the study of the interaction between PTN and a series of CS-E tetrasaccharides with additional aromatics rings in their basic structure. Using CSP, we determined that the tetrasaccharide model of CS-E, **1**, and its mimetic with two extra aromatics rings, **2**, bind to the same region of PTN and that **3** probably due to aggregation or unfolding cannot be used in the same chemical shifts perturbation analysis. From this analysis it can be concluded that a tetrasaccharide with adequate terminal residues has the minimum length for interaction. We also proved that the two regions most affected by the complexation are flexible areas, corresponding to the main point of flexibility: The region between two beta-sheets, according to the NMR structures, can be considered as an IDR (intrinsically disorder region) and a loop between two strands of CTD. 

## Data Availability

Data is contained within the article and Appendix A.

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
