# Peer review of "Pleiotrophin Interaction with Synthetic Glycosaminoglycan Mimetics"

_pharmaceuticals, 2022, doi:10.3390/ph15050496_

Round 1

Reviewer 1 Report

The work from Nieto and co-workers describes the binding of a small family of synthetic polyanionic tetra-saccharides tothe Pleitrophin (PTN) protein. The work follows previous work from the same group (references 6 and 7 of this manuscript), but in the current contribution the binding has been studied from the protein perspective. The study describes a conventional approach with somehow interesting results, considering the importance of the biomolecular system and the scarce number of examples of CS-mimetic small molecules binding to target proteins. The manuscript could become suitable for publication in Pharmaceuticals, but just after corrections and additions are included as follows.

1) First of all, the use of English should be thoroughly revised, since there are several (many) grammar and spelling mistakes in the current version of the draft.

2) The supporting information must be also revised, including a more detailed description on the data included in that Supp Info. Also please double check all the tables, since most of them include parts in Spanish.

3) Scheme 1 would be more informative if the authors use colors to highlight the differences between the 1-3 tetra-saccharides.

4) In general, the manuscript is too descriptive and observational. The authors performed a series of titration experiments and used the CSPs to extract apparent binding constants and potential binding epitopes in the protein. However, little discussion on the conclusions extracted from these values is included.

5) For example, the binding epitopes in the protein are not exploited or described enough. The authors might consider to include some docking or molecular dynamics simulations to support the binding observed experimentally, both in terms of strength and binding sites.

6) The comparison of the Kd with respect to those measured using other techniques (fluorescence polarization titrations, IC50) is rather intriguing. First, in the text, the authors just refer to ref. [6], while ref.[7] (the one included as referee materials) is also very relevant. The differences between techniques is rather large, even considering that IC50 and Kd are not exactly the same parameters. The authors might discuss on that important issue, stating if this is due to different experimental conditions (buffer, temperature, pH, etc.) or any other key factor. If the only difference is the overall concentration, maybe some competing aggregation is playing a role in the NMR measurements.

7) Regarding point 6) I guess the authors used fluorescently labeled derivatives of the saccharides for the fluorescence polarization titration experiments. These labels are organic molecules that can produce additional non-specific interactions with the proteins, thus leading to an apparent stronger binding. Please comment on that.

8) The fitting procedure from the CSP remains unclear to me. Tables 3 and 4 In the supporting information show a long list of different Kd from different residues. Why the authors selected some specific ones and under which criterion?

9) If points 5) to8) are not clear enough, therefore the values of the actual affinities remain unclear. The authors should consider testing the binding with any other alternative experimental technique like ITC or SPR. I understand this manuscript is for a special issue on NMR, but complementary biophysical techniques are extremely useful to discard potential errors or to decide which is the most reliable value of a binding constant, especially when two techniques show dissimilar values like the ones here described.

Author Response

Please find a revised version of the submission after extensive correction. We have used the revision mode available in Word for marked them.

Although was not in this paper, now is better described the way we estimate the binding constants from FPCA (Fluorescence Polarization Competition Assay). and its advantages to other methods to estimate KD.

Figures have been divided and enlarged for better comprehension.

  1. Results, have been reordered, 3. Discussion, is reordered and extended. and now Conclusions has been introduced.

Also Supplementary Materials have been revised and extended.

Referee 1

The work from Nieto and co-workers describes the binding of a small family of synthetic polyanionic tetra-saccharides to the Pleitrophin (PTN) protein. The work follows previous work from the same group (references 6 and 7 of this manuscript), but in the current contribution the binding has been studied from the protein perspective. The study describes a conventional approach with somehow interesting results, considering the importance of the biomolecular system and the scarce number of examples of CS-mimetic small molecules binding to target proteins. The manuscript could become suitable for publication in Pharmaceuticals, but just after corrections and additions are included as follows.

1) First of all, the use of English should be thoroughly revised, since there are several (many) grammar and spelling mistakes in the current version of the draft.

Revised using commercial software.

2) The supporting information must be also revised, including a more detailed description on the data included in that Supp Info. Also please double check all the tables, since most of them include parts in Spanish.

Thank you, it has been revised and corrected.

3) Scheme 1 would be more informative if the authors use colors to highlight the differences between the 1-3 tetra-saccharides.

Thank you for the suggestion. We have done this.

4) In general, the manuscript is too descriptive and observational. The authors performed a series of titration experiments and used the CSPs to extract apparent binding constants and potential binding epitopes in the protein. However, little discussion on the conclusions extracted from these values is included.

We have used a commercial software for processing the data. This software calculates, from the data at growing concentrations of the ligand, Kd, the Chemical Shift Perturbation, and their respective errors, assuming a 1:1 complex. The discussion has been extended.

5) For example, the binding epitopes in the protein are not exploited or described enough. The authors might consider to include some docking or molecular dynamics simulations to support the binding observed experimentally, both in terms of strength and binding sites.

This particular case is too complicated to use both tools properly. We have been trying to have structures representatives of complexes for the case of Midkine and PTN (both proteins constitute the same family). We cannot model the conformational plasticity of the proteins, the range of variation / motion is too large as to be considered by standard tools as MD, or it is out of the time scale of the MD simulations. In preliminary works we use only one conformation of the protein with a cavity in the middle using docking programs, obtaining with many positions/orientations of the ligand along the cavity. Later we have tried to use several models of the protein that represent of the complete range of structures with similar results but complicated by the fact that there is more than one protein structure. They are reported in the latest paper Int. J. Mol. Sci. 2022, 23, 3026

6) The comparison of the Kd with respect to those measured using other techniques (fluorescence polarization titrations, IC50) is rather intriguing. First, in the text, the authors just refer to ref. [6], while ref.[7] (the one included as referee materials) is also very relevant. The differences between techniques is rather large, even considering that IC50 and Kd are not exactly the same parameters. The authors might discuss on that important issue, stating if this is due to different experimental conditions (buffer, temperature, pH, etc.) or any other key factor. If the main difference is the overall concentration, maybe some competing aggregation is playing a role in the NMR measurements.

We now include both references [6] and [7]. We agree with the referee that [7] is more adequate, but we had not this paper accepted at the moment of the submision. This part has been rewritten.

The main difference between both techniques is the concentration that for Fluorescence Polarization Competition Assay is in the range of nano M, while the NMR measurements were 0.5 mM of protein. In addition, to minimise unspecific interactions FPCA is recorded in presence of 0.5% of BSA. Then, the probability of measure aggregates with several ligand molecules is maximised in NMR titration. Briefly, the fluorescence measured comes from a heparin derived fluorescence probe that is displace by the ligand of interest, passing of a high polarization (bound) to a low polarization (free) situation. Such difference is what is detected in an assay using microplates. NMR the detection is based on chemical shift differences between the free and associate estate of the proteins.

7) Regarding point 6) I guess the authors used fluorescently labeled derivatives of the saccharides for the fluorescence polarization titration experiments. These labels are organic molecules that can produce additional non-specific interactions with the proteins, thus leading to an apparent stronger binding. Please comment on that.

The nature of the assay avoids the use of derivatives of the molecules measured. The binding is measured with the molecules used for the 15N-HSQC titration.

8) The fitting procedure from the CSP remains unclear to me. Tables 3 and 4 In the supporting information show a long list of different Kd from different residues. Why the authors selected some specific ones and under which criterion?

The main criterium was a good fitting performance in both cases 1 and 2 close to the places of interaction: hinge and interstrand loop and with large Chemical Shifts Variations. We haven´t found many cases of good fitting curves for both compounds. In the previous examples with hexasaccharides by Wang, the Kd were concentrated into two values one for each domain CTD and NTD. Here we have found a rather disperse response.

We have assume that part of the response to CSP is contaminated by the effect of ring currents.

9) If points 5) to8) are not clear enough, therefore the values of the actual affinities remain unclear. The authors should consider testing the binding with any other alternative experimental technique like ITC or SPR. I understand this manuscript is for a special issue on NMR, but complementary biophysical techniques are extremely useful to discard potential errors or to decide which is the most reliable value of a binding constant, especially when two techniques show dissimilar values like the ones here described.

ITC is a quite expensive technique, in terms of protein. In this case it is needed to have a solution of protein that is titrated with the ligand. The consumption of protein is worse, for high quality measurements using 1mL volume cells, it will be equivalent to NMR in terms of stoichiometry.

SPR needs a previous optimization as the tetrasacharides are too short for the attachment to the surface and the protein to perform the experiments in competition mode. Protein in presence of variable concentrations of the ligands are flowed and the response is observed. Nevertheless, the results could be also different as they are in superficial arrangement which is different than a solution one.

We are quite confident with the Fluorescence Polarization Competition Assay performance.

We have a good description in reference 7.

In addition, we have revised all the submission. The changes are marked with the function 

Reviewer 2 Report

The paper report results about an interesting theme. And the experimental scheme adopted is properly applied. Nevertheless to my opinion the paper cannot be accepted in the present form. The data reported are not correctly reported and commented.

Particularly the NMR study is not adequately presented. The 15N-HSQC spectra are too small and visible  with difficulty. Fig 1 a) and c) at a detailed comparison appear absolutely identical also in detail to the a) and b) of Fig.2. Particularly in the Fig.2   c there are not differences in CS displacements with respect to 1c. Neither those  of low intensity  usually find in structured regions  nor more large displaced ones  in regions poorly structured ( or IDP) that are present as cited in the Introduction. Particularly the color code in Fig.s 1c and  2c has been neglected. Moreover due to the differences found between 1 and 2 in Kd it is obvious that differences in the interaction in the early stages of the interaction may occur between PTN and either 1 or 2.

 I think that when the experimental results have been obtained as written be the AAS as in this case should be reported in detail and commented. Thus the results in form of stick of the 15-N HSQC of the several additions from 0.0 to 3.0  of 1 and  after of 2 made to PTN should be reported and adequately commented. In this context the quality of the Fig.s is very important . In fact the same paper reports important differences in the Kd of PTN with 1 or 2. The differences in the early stages of the interaction may help also to evidenziate the role of the O-bz on the structure of 2.  The labels  in Fig. 1 and 2 are totally invisible. The structure should be described larger than now in a separate Fig. It would be interesting to report also  the position of the W in the structure because the CS  of its NH  it clearly involved in the binding both with 1 and 2. Never cited. Few attention has been attributed to the distribution of the polar residues along the chain of PTN which could be the primary binding sites toward the sulfate group of 1 and 2.

In this case the paragraph of the ref 12 ( Williamson) about the difference of the CS in the protein interactions would help more. In some cases a separate analysis may give interesting indications. Differences in the 1-H CS and the 15-N chemical shifts of proteins may lead to different  conclusions in the study of the interactions.  The standard algorythm for the HSQC may result in a loss of information. NMR study of labelled proteins are a high cost research ( instruments and isotope labelling ) thus I believe that all the results should be used at the best.

In conclusion the paper should use all the results only re-elaborating those  already obtained to reach better conclusions. The Figures should indicate t the reader the point discussed in the results.

The last suggestion is about the insertion of the description of the structure of PTN as reported in the paper submitted and enclosed to this. I believe that should be also reported here in the introduction to give to the reader an idea of the PTN protein under study.

With these results the AAs say that :  

Thus  I think that the results should be more detailed to convince about the conclusions cited at the end of the paper. In the

“We hypothesize that the central region of PTN is an Intrinsically Disordered Region (IDR) and could

modify its properties upon binding. The second tetrasaccharide has two benzyl groups and shows

similar effects on PTN”. 

Author Response

Please find a revised version of the submission after extensive correction. We have used the revision mode available in Word for marked them.

Although was not in this paper, now is better described the way we estimate the binding constants from FPCA (Fluorescence Polarization Competition Assay). and its advantages to other methods to estimate KD.

Figures have been divided and enlarged for better comprehension.

  1. Results, have been reordered, 3. Discussion, is reordered and extended. and now Conclusions has been introduced.

Also Supplementary Materials have been revised and extended.

Referee 2

The paper report results about an interesting theme. And the experimental scheme adopted is properly applied. Nevertheless to my opinion the paper cannot be accepted in the present form. The data reported are not correctly reported and commented.

Particularly the NMR study is not adequately presented. The 15N-HSQC spectra are too small and visible  with difficulty. Fig 1 a) and c) at a detailed comparison appear absolutely identical also in detail to the a) and b) of Fig.2. Particularly in the Fig.2   c there are not differences in CS displacements with respect to 1c. Neither those  of low intensity  usually find in structured regions  nor more large displaced ones  in regions poorly structured ( or IDP) that are present as cited in the Introduction. Particularly the color code in Fig.s 1c and  2c has been neglected. Moreover due to the differences found between 1 and 2 in Kd it is obvious that differences in the interaction in the early stages of the interaction may occur between PTN and either 1 or 2.

I apologize by mistake Figure 1 was load twice as Figure 2.

 I think that when the experimental results have been obtained as written be the AAS as in this case should be reported in detail and commented. Thus the results in form of stick of the 15-N HSQC of the several additions from 0.0 to 3.0 of 1 and after of 2 made to PTN should be reported and adequately commented. In this context the quality of the Fig.s is very important . In fact the same paper reports important differences in the Kd of PTN with 1 or 2. The differences in the early stages of the interaction may help also to evidenziate the role of the O-bz on the structure of 2. The labels in Fig. 1 and 2 are totally invisible. The structure should be described larger than now in a separate Fig. It would be interesting to report also the position of the W in the structure because the CS of its NH it clearly involved in the binding both with 1 and 2. Never cited. Few attention has been attributed to the distribution of the polar residues along the chain of PTN which could be the primary binding sites toward the sulfate group of 1 and 2.

We are sorry about the quality of the figures. They have been improved by dividing them into three larger figures.

The influence of the substituents in the structure of the tetrasaccharides have been reported previously (ref 6) They do not change along the series in spite of the expected steric tension. By Transfer-NOESY experiments we also have determined that the 3D structure do not change on binding (ref 5)

There are 4 W residues but only one, in position 59 is influenced by binding. The other three (18, 20 and 74) were detected but did not pass the threshold to be considered influenced by the binding of 1 and 2. This is a clear prove that this type of binding in not as the standard lectin ones.

As we mentioned in ref. 6 the positive residues in the beta sheets are pointing towards the inner cavity formed by the two structured domains in structure 1 of the 10 structures deposited in the Protein Data Bank and in the BioMagResBank. Many of the residues pointing towards the internal cavity are charged and affected by the addition of the ligands.

In this case the paragraph of the ref 12 ( Williamson) about the difference of the CS in the protein interactions would help more. In some cases a separate analysis may give interesting indications. Differences in the 1-H CS and the 15-N chemical shifts of proteins may lead to different conclusions in the study of the interactions. The standard algorythm for the HSQC may result in a loss of information. NMR study of labelled proteins are a high cost research (instruments and isotope labelling ) thus I believe that all the results should be used at the best.

In conclusion the paper should use all the results only re-elaborating those already obtained to reach better conclusions. The Figures should indicate t the reader the point discussed in the results.

The last suggestion is about the insertion of the description of the structure of PTN as reported in the paper submitted and enclosed to this. I believe that should be also reported here in the introduction to give to the reader an idea of the PTN protein under study.

We have already included the following description of the structure:

The PTN structure has two pseudotrombospondin type 1 repeat domains (TSR) stabilized by disulfide bonds, linked by a central disordered region, and with both ends also unstructured. Basic amino acids predominate in both ends and both beta-sheet domains. They can be grouped in clusters. In the NTD, R35, R39, R49, R52, and K54 are in the same face, the internal one. The CTD residues K84, R86, K107, and K68, K91, and R92, classified as clusters 1 and 2, respectively, correspond to the region close to the central linker and the loop between strands 2 and 3 of CTD. Some of the structures deposited in the protein data bank clusters 1 and 2 are spatially close. There is another group of basic residues on CTD in strands 2 and 3, K84, R86, and K107, directed towards the internal part of PTN in some of the structures[9, 13]

We think that the structure of the PTN is adequately described.

With these results the AAs say that :  

Thus  I think that the results should be more detailed to convince about the conclusions cited at the end of the paper. In the

“We hypothesize that the central region of PTN is an Intrinsically Disordered Region (IDR) and could modify its properties upon binding. The second tetrasaccharide has two benzyl groups and shows similar effects on PTN”.

We have extended the discussion and conclusion parts

Reviewer 3 Report

Pleiotrophin (PTN), known as one of the heparin-binding proteins, plays an important role in the embryogenesis of the central nervous system. It is also involved in other physiological processes such as tissue repair, cell differentiation, inflammation, angiogenesis and cancers. PTN is, therefore, related to various diseases, having caught large interests as a drug target. Although the structures of PTN are available, the binding site for chondroitin sulfate (CS) is not known. The authors synthesized three kinds of tetrasaccharides, one of which was a short CS-E and the other two of which were its derivatives, and they studied the interaction of each with 15N-labeled PTN using NMR. Practically, they titrated each of the three compounds into [15N]-PTN to observe the chemical shift perturbation of amide 1H/15N groups of PTN. As the result, they found that at least one of them, tetrasaccharide (2), interacted with loop regions of PTN, which matched the regions that were also suggested as the interaction sites with the original CS-E mimic, namely, tetrasaccharide (1). The authors have thus concluded that these tetrasaccharides well mimic heparin when they interact with PTN.

There are several minor concerns that the authors should address before publication.

The abbreviated word “GAG” first appears in the manuscript before the original term “glycosaminoglycan”.

in page 2

The word “alfa” should be “alpha”.

In the interpretation of the spectra upon the titration of the compound (3) into PTN, I did not understand what “decomposition” in the text means (probably “unfolding”?). Decomposition may give an impression that PTN was fractionated into small fragments. Although the following is based on my speculation, I think that the interaction observed was in an intermediate exchange mode, which usually deletes most peaks, resonating above 8.6 ppm, through an exchange broadening mechanism (Rex) and which leaves only peaks with high intensities ,resonating around 7.0-8.5 ppm, including the peaks from Gln and Asn sidechains. Alternatively, the compound (3) may have interacted with other regions of PTN non-specifically, and inducing an aggregation and the subsequent precipitation of the mixtures of PTN and the compound (3). In such cases, precipitation does not necessarily mean that the protein was unfolded. In some cases, the protein may aggregate and precipitate while maintaining its conformation, just as seen in isoelectric point precipitation and ammonium sulfate precipitation. Since compound (3) has more aromatic rings than (1) and (2), (3) is more hydrophobic and thus may have acted as a glue between the complex of PTN and (3) in aggregation. Reducing the salt concentration from 150 to 10 mM may cause the molecules to disperse again.

The structure figures (Figs. 1 (b) and 2 (b)) are too simple. On the figures, I failed to locate the binding regions and residues that are described in the manuscript. The figures should include the domain names, important beta sheet numbers, residue numbers (at least those of Fig. 3), N/C terminals, and the PDB accession code, so that readers can easily relate the perturbation plots (c) to the conformations (b).

Are figures 1 and 2 not the same (due to a mistake in pasting the original figures)?

p. 4 “suggesting a larger interaction”

Does it not actually mean a “stronger” interaction? Readers may interpret "Large interaction" as large interaction areas.

The manuscript does not describe how the authors prepared the 15N-labeled PTN. Did the authors use an E. coli expression system? How did they purify the protein?

In Materials and Methods,

“0.5 mM 15N-pleiotrophin in 300 mL” should be “…300 micro L” instead of “mL”.

Author Response

Please find a revised version of the submission after extensive correction. We have used the revision mode available in Word for marked them.

Although was not in this paper, now is better described the way we estimate the binding constants from FPCA (Fluorescence Polarization Competition Assay). and its advantages to other methods to estimate KD.

Figures have been divided and enlarged for better comprehension.

  1. Results, have been reordered, 3. Discussion, is reordered and extended. and now Conclusions has been introduced.

Also Supplementary Materials have been revised and extended.

Referee 3

Pleiotrophin (PTN), known as one of the heparin-binding proteins, plays an important role in the embryogenesis of the central nervous system. It is also involved in other physiological processes such as tissue repair, cell differentiation, inflammation, angiogenesis and cancers. PTN is, therefore, related to various diseases, having caught large interests as a drug target. Although the structures of PTN are available, the binding site for chondroitin sulfate (CS) is not known. The authors synthesized three kinds of tetrasaccharides, one of which was a short CS-E and the other two of which were its derivatives, and they studied the interaction of each with 15N-labeled PTN using NMR. Practically, they titrated each of the three compounds into [15N]-PTN to observe the chemical shift perturbation of amide 1H/15N groups of PTN. As the result, they found that at least one of them, tetrasaccharide (2), interacted with loop regions of PTN, which matched the regions that were also suggested as the interaction sites with the original CS-E mimic, namely, tetrasaccharide (1). The authors have thus concluded that these tetrasaccharides well mimic heparin when they interact with PTN.

There are several minor concerns that the authors should address before publication.

The abbreviated word “GAG” first appears in the manuscript before the original term “glycosaminoglycan”.

Thank you we have corrected it

in page 2

The word “alfa” should be “alpha”.

Thanks, it is corrected.

In the interpretation of the spectra upon the titration of the compound (3) into PTN, I did not understand what “decomposition” in the text means (probably “unfolding”?). Decomposition may give an impression that PTN was fractionated into small fragments. Although the following is based on my speculation, I think that the interaction observed was in an intermediate exchange mode, which usually deletes most peaks, resonating above 8.6 ppm, through an exchange broadening mechanism (Rex) and which leaves only peaks with high intensities,resonating around 7.0-8.5 ppm, including the peaks from Gln and Asn sidechains. Alternatively, the compound (3) may have interacted with other regions of PTN non-specifically, and inducing an aggregation and the subsequent precipitation of the mixtures of PTN and the compound (3). In such cases, precipitation does not necessarily mean that the protein was unfolded. In some cases, the protein may aggregate and precipitate while maintaining its conformation, just as seen in isoelectric point precipitation and ammonium sulfate precipitation. Since compound (3) has more aromatic rings than (1) and (2), (3) is more hydrophobic and thus may have acted as a glue between the complex of PTN and (3) in aggregation. Reducing the salt concentration from 150 to 10 mM may cause the molecules to disperse again.

We want to thank the referee 3 for its clever comments. However, regarding the observation in figure 2a the superimposition of traces seems independent on exchange the peaks can be superimposed and only the intensity varies. We have seen this effect previously in the interaction with MK when used a compound similar to 3, but complementary sequence. This compound precipitates in STD measurements (defect of protein) but the complex was soluble in absence of NaCl. Unfortunately, it is difficult to implement experimentally as we perform the titration by dissolving the amount required in the solution of the previous point, and it will be impossible do the first point (no NaCl). We will consider the effects of aggregation in future.

The structure figures (Figs. 1 (b) and 2 (b)) are too simple. On the figures, I failed to locate the binding regions and residues that are described in the manuscript. The figures should include the domain names, important beta sheet numbers, residue numbers (at least those of Fig. 3), N/C terminals, and the PDB accession code, so that readers can easily relate the perturbation plots (c) to the conformations (b).

These features have been modified. The Beta sheets positions are at the top of the CSP graphics.

The comments of the referee have been implemented in the text.

Are figures 1 and 2 not the same (due to a mistake in pasting the original figures)?

Yes, they were. That mistake has been corrected

  1. 4 “suggesting a larger interaction”

Does it not actually mean a “stronger” interaction? Readers may interpret "Large interaction" as large interaction areas.

Thank you.

The manuscript does not describe how the authors prepared the 15N-labeled PTN. Did the authors use an E. coli expression system? How did they purify the protein?

The reference has been added to the text. Was already cited

In Materials and Methods,

“0.5 mM 15N-pleiotrophin in 300 mL” should be “…300 micro L” instead of “mL”.

Yes, thank you for the correction.

Round 2

Reviewer 1 Report

The authors have addressed all my main concerns, rendering an improved version of the manuscript. This is a very good contribution to the journal.

Author Response

Thanks for the improvements done to our manuscript. 

Reviewer 2 Report

The paper is rather better now.

Nevertheless some concern remain to be corrected. The use of the term aggregation for the compound 3 seems to me inappropriate.

In the three sentences reported here I think should be considered the possibility of a tight binding that usually leads to the disappearence of the 15N amide ( 15). In fact when the CS changes too much  a huge broadening occurs and , eventually a new peaks appears at very high protein ligand ratio ,( usualy not reached for the lack of material or precipitation). This argument would  lead to consider the 3 as the most strong binder of PTN protein.

Thus I suggest to change the term “aggregation” and to insert these consideration in the following three sentences:

  1. In the case of 3, the protein peaks disappeared along with the titration. As a result, the precise exchange regime is unclear; it may be the consequence of a combination of slow exchange followed by aggregation or another process that dismiss the amount of observable protein in solution (Fig. 2). A similar ligand-induced signal decrease was seen in the titration of PTN by Heparin reported by Rauvala[13]. In both cases, the final results are similar, as shown in figure 2, with signals that can be attributed to unfolded structures observed.

  1. Almost all backbone amide peaks can be observed and assigned, but not all showed changes in the chemical shifts. 3 behaves differently; in the first titration points, the intensity of the signals decreases, and peaks disappear, but they do not reappear in the complex position as can be expected for a slow chemical shift equilibrium. As an alternative mechanism, aggregation is a plausible hypothesis.

  1. 3) Using CSP, we have determined that the tetrasaccharide model of CS-E, 1, and its mimetic with two extra Pharmaceuticals 2021, 14, x FOR PEER REVIEW 10 of 11 aromatics rings, 2, bind to the same region of PTN and that 3 probably by aggregation cannot be used in the same Chemical Shifts Perturbation Analysis.

TEXT. “Figure 3 displays the CSP of 1 and 2 along the chain, and in the top are the secondary structure elements, beta-strands, and the NTD and CTD region indicated at the same scale. The results obtained for tetrasaccharides 1 and 2 were similar to previous data on CS-E (dp6), CS-A (dp8)[9], Heparin (dp6), CS-E (dp6), DS (dp8), and CS-A (dp8)[8]. The global CSP profile of both compounds is very similar. The magnitude of the variation is clearly more significant for 2, likely for the influence of the aromatic rings of the O-benzyl groups (see figure 3)”.

And “When the secondary structure is considered, there are two regions where the larger CSP are concentrated: the hinge region and the connection between the 2nd and 3rd beta-strands of the CTD. These two regions are close in the space in some crystallographic structures, see figure 4. Interestingly, the CSP magnitudes are very different for both compounds.“

 In this sentences   I believe that  the term “ very similar” is not complertely correct. I think that there is a similarity but “The magnitude of the variation is clearly more significant for 2, likely for the influence of the aromatic rings”  . Inspection of the corrected Fig. s 1 and 2 lead to observe a more marked preference for the loops  than for the beta strands. About the term “larger”  it is difficult to be simply interrpreted. In fact this observation  can be interpreted as an effect of the binding on the disordered regions where the binding leads to a more reduced number of conformation and then  chemical shifts  than that on the structured regions obviously more stable. I am not sure that the analysis of the absolute values of the CS  changes is the better way to understand the binding preferences.

Finally I return to suggestion I made  the AAS to better analyze the data of the binding to PTN of 1 and 2. Reporting in sticks the values of the CS variation upon addition of 1 and 2  to PTN  ( see “molasr ratios s from 0.0 to 3.0-4.0”)  I believe that a graphical comparison of the growth of the CSP upon the addition of 1 or 2  may help in the interpretation of the results much better than the color codes us3ed in the structtures of the  fig.s .

Author Response

1.- Regarding the aggregation issue:

This is the end of a long project and maybe we have not correctly written the paper in order to reflect previous data.

The case of 3 is paradigmatic. We did not measure by transient NMR the interaction of 3 with PTN but we did it with Midkine (reference 6). Both are very similar proteins in structure, function and display similar affinities and binding tendencies. By FPCA (low concentration and precise binding constants) the magnitude of binding constant was 1 > 2 > 3 with nearly one order of magnitude step between them. Later we reproduce the experiments with PTN, except the compound 3, due to lack of compound. 1 and 2 behaved similarly with similar IC50 and increase of binding when aromatics were introduced for MK and PTN (5,6). Thus we have assumed that the binding to NK and PTN is similar. This is reinforced by the behavior of other compounds with complementary sequence that have similar performance studied in ref 6 in presence of MK.

The 15N-HSQC reported here are the only equimolecular experiments performed at high protein concentration (0.5mM) for these tetrasaccharides for both MK and PTN (we have no access to 15N labelled MK).

On the other hand, it is true that the fully substituted tetrasaccharide in the previous experiments with MK showed a borderline behavior with respect to the fast or slow binding equilibria. When we did the Transfer-NOESY experiments at the standard concentrations, 1.0 – 0.5 mM, the signals broader and a large spin diffusion was observed due to larger size assemblies but still with single signals for each proton. This behavior disappears with dilution (now narrow signals corresponding to 1:1 complex) reflected in NOESY, and we performed the rest of the experiments at 900MHz and lower concentration (.25uM). See ref 6.

Then this observation makes us to use the term aggregation in this paper because similar 1:1 complex are well observed in fast exchange regime for 1 and 2, but not for 3 that is compatible with shorter correlation times, larger molecular sizes and precipitation.

As we do not observe the appearance of the complex signals at the end of the titration, we have decided not classify the behavior as it cannot distinguished between the unfolding or aggregation. In addition, the beta-sheets domains, that are the only secondary structure elements, are stabilized by disulfide bridges, there are not others elements of secondary structure. An unfolding process that implies the breaking of disulfide bridges is not likely to happen in these conditions.

  • We have removed the last sentence of the paragraph “Figure 3. ..

“The magnitude of the variation is clearly more significant for 2, likely for the influence of the aromatic rings of the O-benzyl groups (see figure 3)”

  • We have modified the sentence for a more precise one:

“When the secondary structure is considered, there are two regions where the larger relative CSP values are concentrated: the hinge region and the connection between the 2nd and 3rd beta-strands of the CTD.”

  • Please find that the correction in the caption of Figure 2, may correct the last misunderstanding.